# Short- and Long-Term Effects of Suboptimal Selenium Intake and Developmental Lead Exposure on Behavior and Hippocampal Glutamate Receptors in a Rat Model

**DOI:** 10.3390/nu14163269

**Published:** 2022-08-10

**Authors:** Anna Maria Tartaglione, Melania Maria Serafini, Francesca Ferraris, Andrea Raggi, Annalisa Mirabello, Rita Di Benedetto, Laura Ricceri, Miriam Midali, Francesco Cubadda, Luisa Minghetti, Barbara Viviani, Gemma Calamandrei

**Affiliations:** 1Centre for Behavioral Sciences and Mental Health, Istituto Superiore di Sanità (ISS), 00161 Rome, RM, Italy; 2Department of Pharmacological and Biomolecular Sciences, Università degli Studi di Milano, 20126 Milan, ML, Italy; 3Department of Food Safety, Nutrition and Veterinary Public Health, Istituto Superiore di Sanità (ISS), 00161 Rome, RM, Italy; 4Research Coordination and Support Service, Istituto Superiore di Sanità (ISS), 00161 Rome, RM, Italy

**Keywords:** selenium, lead, behavior, glutamatergic receptors, perinatal exposure

## Abstract

Selenium (Se) is an essential trace element required for normal development as well as to counteract the adverse effects of environmental stressors. Conditions of low Se intake are present in some European countries. Our aim was to investigate the short- and long-term effects of early-life low Se supply on behavior and synaptic plasticity with a focus on the hippocampus, considering both suboptimal Se intake per se and its interaction with developmental exposure to lead (Pb). We established an animal model of Se restriction and low Pb exposure; female rats fed with an optimal (0.15 mg/kg) or suboptimal (0.04 mg/kg) Se diet were exposed from one month pre-mating until the end of lactation to 12.5 µg/mL Pb via drinking water. In rat offspring, the assessment of motor, emotional, and cognitive endpoints at different life stages were complemented by the evaluation of the expression and synaptic distribution of NMDA and AMPA receptor subunits at post-natal day (PND) 23 and 70 in the hippocampus. Suboptimal Se intake delayed the achievement of developmental milestones and induced early and long-term alterations in motor and emotional abilities. Behavioral alterations were mirrored by a drop in the expression of the majority of NMDA and AMPA receptor subunits analyzed at PND 23. The suboptimal Se status co-occurring with Pb exposure induced a transient body weight increase and persistent anxiety-like behavior. From the molecular point of view, we observed hippocampal alterations in NMDA (Glun2B and GluN1) and AMPA receptor subunit trafficking to the post-synapse in male rats only. Our study provides evidence of potential Se interactions with Pb in the developing brain.

## 1. Introduction

Early-life adverse environmental factors, including nutritional deficiencies and exposure to chemical contaminants, may have detrimental effects on the developing brain, contributing to the risk of neurodevelopmental and neurodegenerative disorders.

In particular, research in animal models and human populations shows the critical impact of the lack of specific micronutrients, such as selenium (Se), during pregnancy and the first years of life on the maturation of psychomotor and cognitive functions [1].

Se is an essential element for humans and animals as it is needed for the synthesis of selenoproteins and is found as L-selenocysteine in the active center of several selenoprotein enzymes, such as glutathione peroxidases (GPx), iodothyronine deiodinases (DIO), and thioredoxin reductases (TrxR), which have a pivotal role in antioxidant defenses, reactive oxygen species (ROS) scavenging, and thyroid hormones metabolism [2,3,4]. Thanks to a specific Se hierarchy in the human body, the brain maintains relatively high selenoprotein expression even in the presence of low dietary supply [5], suggesting a relevant role for Se in this organ. Specifically, brain Se is maintained at the expense of other tissues [6], and the hippocampus is one of the CNS regions with higher Se levels [7]. This task is accomplished by the expression of specific transporters in the brain, among which low-density lipoprotein-receptor-related protein 8 (LRP8, also known as ApoER2) is the most prominent [8,9]. LRP8 is located at the blood–brain barrier (BBB) and is expressed by astrocytes and neurons; it facilitates the uptake of Se conveyed by selenoprotein P (SEPP1), the key selenoprotein for Se transport, supply to tissues and Se homeostasis in the whole body [9]. Although SEPP1 is mainly produced by the liver, the intracellular expression of this protein has been reported for neurons [10] and astrocytes [11,12], playing a role in brain Se hierarchy.

Notably, the hippocampus is more dependent on SEPP1 for optimal Se concentration than other brain regions [13]. However, the exact role of Se and selenoproteins in neurodevelopmental processes remains unclear and little is known about the possible involvement of Se in synaptic spines formation, maturation, and stabilization in the hippocampus.

Human cohort studies have revealed an association between the maternal Se status and children’s neurobehavioral development. A positive effect of maternal selenium status on children’s language and psychomotor development at 1.5 years of age has been reported [14]. In a Polish cohort study, a significant positive association was detected between Se levels in the blood collected during the first trimester of pregnancy, child motor skills at 1 and 2 years of age, and cognitive development at 2 years of age [15,16]. Recent data show that both low and high levels of cord serum Se have adverse effects on infant neuropsychological development [17], suggesting that an intermediate range of Se intake during pregnancy might favor brain and behavior development. This is consistent with the generally recognized U-shaped relationship between Se and health, with adverse health effects associated with, or caused by, both Se deficiency and excess [18]. However, the specific mechanisms behind the positive effect of Se on brain and behavior development have yet not been clarified. Notably, in experimental studies, the nutritional deficiency of Se is accompanied by decreased GPx activity in the brain associated with the altered turnover of specific neurotransmitters [19] and thyroid dysfunction [20]. Although the neurobiological mechanisms by which Se promotes neurodevelopment are not fully understood, both the modulatory effects of Se on thyroid function and the protection of developing neural cells from oxidative damage could exert beneficial effects on cognitive and emotional maturation [21,22].

An open research question is whether suboptimal Se intake, which occurs in many populations worldwide, could harm neurodevelopment exacerbated by the presence of adverse environmental factors to which the fetus and the infant may be exposed. To date, the available studies point out the protective role of the Se supply against the detrimental effect of developmental exposure to heavy metals such as methylmercury and lead (Pb). Specifically, the experimental evidence supports Se protective effects on Pb-induced neurotoxicity, spatial learning, and memory deficits induced by Pb exposure at different developmental stages [23,24]. Pb is widely recognized as a central neurotoxin that is primarily critical during pregnancy and early childhood. Low-level environmental Pb exposure impairs neurodevelopment and cognitive functions, especially in infants and small children [25], affecting cortical volumes and cognitive test scores [26]. A benchmark dose lower confidence limit (BMDL) of 1% extra risk equal to 1.2 μg Pb/dL has been identified by the EFSA as a reference point for the risk characterization of Pb when assessing the risk of intellectual deficits in children up to the age of seven years measured by the Full-Scale IQ score [27]. Conditions of suboptimal or low Se intake are present in some European countries, in particular in Eastern Europe [28], and research on whether this may result in increased vulnerability to the neurotoxic action of widespread environmental chemicals, such as Pb, is needed. However, the link between Se status in pregnancy, early exposure to chemical stressors such as Pb, and child neuropsychological development is difficult to ascertain in large birth cohort studies. In this respect, experimental models are extremely useful to disentangle the interaction between micronutrients and toxicants in critical brain developmental windows.

We previously demonstrated that developmental Pb exposure during gestation and lactation until weaning affected glutamatergic receptor expression and distribution to the post-synaptic site in a sex-dependent fashion. Selected behavioral domains were altered, with more pronounced effects in female rats exposed to Pb [29]. The glutamatergic system plays a crucial role in early life since the distribution of receptors at the synapse, their trafficking, and the specific balance among different subunits contribute defining excitatory synaptic connectivity.

The present study was designed to examine the potential interactions between suboptimal dietary Se and low-level Pb exposure on different behavioral domains and the synaptic organization of glutamatergic ionotropic receptors in the hippocampus.

To this end, based on the results of a previous developmental study, indicating significant effects of suboptimal Se intake on the behavioral and inflammatory responses [30], we established an animal model of Se restriction and developmental Pb exposure: female rats fed on a diet with optimal (0.15 mg/kg) or suboptimal (0.04 mg/kg) Se content (as L-selenomethionine, the main Se chemical species in the diet) were exposed, for one month pre-mating until the end of lactation, to Pb in drinking water (at 12.5 µg/mL Pb). Motor, emotional, and cognitive endpoints were assessed at different life stages in the offspring and complemented by the evaluation of the hippocampal glutamatergic machinery. In particular, we measured the expression and synaptic distribution of the main N-methyl-D-Aspartate receptor (NMDA) and α-amino-3-hydroxy-5-methyl-4-isoxazolepropionic acid (AMPA) receptor subunits (GluN2A, GluN2B, GluN1, GluA1, and GluA2), together with postsynaptic density protein 95 (PSD-95) at PND 23 and 70. In addition, we evaluated the expression of LRP8 and SEPP1 to monitor the capability of the brain to uptake Se in different experimental conditions.

## 2. Materials and Methods

### 2.1. Animals and Experimental Design

Wistar rats were kept under standard animal housing (temperature 20 ± 2 °C; humidity 60–70%) with food and water ad libitum, under a 12 h–12 h light/dark cycle (lights on from 7:00 a.m. till 7:00 p.m.). Following the adaption period and 8 weeks before breeding, females (weighing 250 ± 25 g) were assigned to one of the two experimental groups with either optimal or suboptimal dietary Se intake (see ‘Diets’). After the first 4 weeks under this dietary regimen, the rats were additionally given water containing either 0 (Vehicle, Veh) or 12.5 µg/mL Pb through pregnancy and lactation until weaning (PND 23) in their drinking bottles. Thus, the resulting four experimental groups were Se Opt/Veh, Se Opt/Pb, Se Subopt/Veh, and Se Subopt/Pb.

Female rats were mated with males (2:1) for 4 to 5 days to cover the duration of an estrous cycle. The day of birth was designated as PND 0. At PND 1, the litters were culled to equal numbers to standardize litter size to have ten pups per litter. At weaning (PND 23) male and female offspring were separated and housed at 4 per cage until the end of all experiments. The offspring were fed with the dietary regimen received by their respective dams. The animals were regularly weighed, and their food and water consumption was recorded every other day.

The timeline of the experimental design is reported in Figure 1.

### 2.2. Se-Containing Diets

Rat diets were prepared by Envigo by supplementing a selenium-deficient diet (Teklad Custom Diet TD.92163), containing <0.01 mg Se/kg with L-selenomethionine to obtain a diet with optimal (Opt, 0.15 mg Se/kg) or suboptimal (Subopt, 0.04 Se mg/kg) Se content, respectively.

### 2.3. Pb Exposure

Pb acetate (Pb(Ac)2·3H2O) was purchased from Sigma Aldrich (Merck KGaA, Darmstadt, Germany). Test solutions for animal treatment were prepared in soft tap water, slightly acidified (pH 4.6) with acetic acid and containing no Pb acetate (Veh) or Pb acetate at a concentration of 12.5 µg/mL of Pb (as element). The selected medium ensured the complete dissolution of the Pb salt, and acceptance by the animals was the same as plain tap water. The Pb concentration in the test solutions was analytically checked, and stability was verified for 5 days. The solutions were prepared fresh every 3–4 days and used as drinking water for animal treatment. The resulting exposure was ~1.25 mg/kg body weight Pb per day.

### 2.4. Plasma and Milk Se Determinations

Blood samples were collected in EDTA (0.2 mg/100 mL) from the left atrium of the hearts of the rats anesthetized using isoflurane. Samples were centrifuged at 3000 rpm for 10 min at 4 °C and plasma was separated, aliquoted, and stored at −80 °C until analysis. Maternal milk was sampled from the offspring’s stomachs at birth (PND 1) and also stored at −80 °C. The sample treatment was performed in a clean room laboratory. The quantitative analysis of the total Se was carried out with external calibration by means of an 8800 Triple Quad mass spectrometer (ICP-MS) from Agilent Technologies (Tokyo, Japan). Plasma was diluted 1:50 with 0.5% *v/v* HNO_3_, and milk was digested in a microwave system. Oxygen was used as reaction gas for mass shift (SeO+) in ICP-MS/MS determinations, and m/z 94 and 96 were resorted to as analytical masses. The accuracy was checked by the reference materials Seronorm Serum L1, NIST 1549 (Non-Fat Milk Powder), and BCR 063R (Skin Milk). A detailed description of the analytical procedures is given in the Appendix A.

### 2.5. Blood Pb Determinations

Blood samples were collected as described in Section 2.4 and stored at −80 °C until analysis. The sample treatment was performed in a clean room laboratory. Blood was diluted 1:50 with 0.2% *v/v* HNO_3_ and 0.005% *v/v* Triton X-100. Procedural blanks submitted to the same sample handling of blood specimens were prepared and their concentration subtracted from that of study samples. The quantitative analysis of total Pb was carried out with the standard addition method in the same instrumental conditions given in 2.4. The accuracy was checked by the control material Seronorm Whole Blood L1. A detailed description of the analytical procedures is given in the Appendix A.

### 2.6. Behavioral Testing

Neonatal stage: One male and one female offspring derived whenever possible from different litters of the four experimental groups (Se Opt/Veh = 10 males, M, 10 females, F; Se Opt/Pb = 9 M, 9 F; Se Subopt/Veh = 10 M, 10 F; Se Subopt/Pb = 9 M, 9 F) underwent the neurodevelopmental test battery described below, including somatic, motor, and sensorial assessment on PND 4, 7, 10, 12, and 13.

For identification purposes, on PND 4, pups were marked on the paw with animal tattoo ink (Ketchum permanent Tattoo Inks green paste, Ketchum Manufacturing Inc., Brockville, ON, Canada). Juvenile/adult stage: rats belonging to different litters (Se Opt/Veh = 10 M, 10 F; Se Opt/Pb = 10 M, 10 F; Se Subopt/Veh = 10 M, 10 F; Se Subopt/Pb = 10 M, 10 F), not subjected to neurodevelopmental test battery, were assessed in the following behavioral tests: Open-Field (OF, PND 30), Spontaneous alternation (Y-maze, PND 35), Elevated Plus Maze (EPM, PND 60), Novel Object Recognition (NOR, PND 63–65), and Morris Water Maze (MWM, PND 68–72). All apparatuses were cleaned with 70% alcohol following each animal testing. All behavioral procedures were carried out between 9:00 a.m. and 3:00 p.m.

#### 2.6.1. Analysis of Ultrasonic Vocalizations (USVs) and Spontaneous Movements in Pups (PND 4–12)

On each day of testing (PND 4, 7, 10 ,and 12), a single pup was placed into an empty glass container (diameter 5 cm; height 10 cm) located inside a sound-attenuating Styrofoam box, to record USVs with simultaneous spontaneous motor activity during a 3 min test as described in Tartaglione et al. (2019; 2020). The number of calls was analyzed by Avisoft SAS Lab Pro (Avisoft Bioacoustics, Berlin, Germany). In accordance with previous studies focused on neonatal rat behavior [29], the following motor patterns were scored: locomotion (general translocation of the body of at least 1 cm in the glass container), immobility, head rising (rising of the head up and forward), face washing (forepaws moving back and forth from the ears to the snout and mouth), wall climbing (alternating forelimb placing movements on the wall of the container), pivoting (locomotor activity involving the front limbs alone and resulting in laterally directed movements), and curling (vigorous side-to-side rolling movements while on the back). The frequency and duration of each behavioral pattern were analyzed using The Observer XT software (version 11, Noldus, Wageningen, The Netherlands).

#### 2.6.2. Somatic Growth and Sensorimotor Development Assessment in Pups (PND 4–12)

At the end of the 3 min recording session, each pup was assessed for reflex development and somatic growth (body weight and length) from PND 4 to 12, as previously described [29].

The righting reflex was assessed by placing the pup on its back over a flat surface and by measuring the time (within 60 s) needed to return to the natural position (all four paws on the floor).

The negative geotaxis was assessed by placing the pup on a 30 cm inclined plane (20°) in a head-down position. The time required to re-orient to a head-up position was recorded (within 60 s).

#### 2.6.3. Nest-Odor Recognition by Homing Test in Pups (PND 13)

To assess early olfactory discrimination, on PND 13, the pups were separated from the dam and kept for 30 min in an incubator (Elmed Ginevri 0GB 1000, Roma, Italy) at 28 ± 1 °C. The apparatus consisted of a grey Plexiglass T-shaped maze (start arm: 25 × 9 cm; choice arms: 12.5 × 9 cm; height: 8.5 cm). Each pup was placed in the start arm and allowed to freely explore the maze for 3 min. The time taken by the pup to reach the goal arm (containing nest litter), the number of entries, and the time spent in the three arms were recorded and analyzed by The Observer XT software (version 11, Noldus, Wageningen, The Netherlands).

#### 2.6.4. Open-Field Test in Juveniles (OF, PND 30)

To assess the locomotor activity during the exploration of a novel environment, rats were tested in the OF test. Rats were individually placed in one corner of the apparatus consisting of a black Plexiglass box (80 × 80 × 60 cm) and video-recorded for 10 min. Distance travelled and mean velocity were analyzed using ANY-Maze software (version 6.35, Stoelting Europe, Dublin, Ireland).

#### 2.6.5. Y-Maze Test in Juveniles (PND 35)

To assess spatial working memory and explorative activity, rats were tested in the Y-maze test. The apparatus consisted of three identical arms (50 × 16 × 32 cm) diverging at a 120° angle one to the other and an equilateral triangular central area. Each rat was placed in the center of the Y-maze and allowed to freely explore the maze for 5 min. The total number of arm entries and a sequential list of arms entered to assess number of alternations were scored registered and analyzed by The Observer XT software (version 11, Noldus, Wageningen, The Netherlands). An alternation was defined as an entry into three different arms on consecutive choices (arm entry = all four paws into an arm). Spontaneous alternation was calculated using the following formula: (number of alterations/total number of entries − 2) × 100. In addition, ethological measures, including frequency and duration scores for open rearing, wall rearing, and grooming, were analyzed.

#### 2.6.6. Elevated Plus Maze in Young Adults (EPM, PND 60)

To assess the anxiety-like behavior, the rats were tested in the EPM based on the natural conflict between the tendency to explore new areas and the avoidance of unsafe ones. The EPM comprised two open arms and two closed arms that extended from a common central platform (Plexiglass black floor and clear walls) and elevated to a height of 60 cm above the floor level. Rats were individually placed on the central platform facing a closed arm and allowed to freely explore the maze for 5 min. The frequencies of the total, open, and closed entries (arm entry = all four paws into an arm) and the time spent in each arm were scored by The Observer XT software (version 11, Noldus, Wageningen, The Netherlands).

#### 2.6.7. Novel Object Recognition in Young Adults (NOR, PND 63–65)

To evaluate the recognition memory of one previously explored object (familiar) compared with one novel object, the rats were tested for NOR. NOR was performed in an OF apparatus (a black Plexiglass box, 80 × 80 × 60 cm). The rats were tested during three sessions (habituation, familiarization, and novelty/retention) with each session lasting 10 min and separated by an interval of 24 h. In the familiarization session, each rat was faced with two identical objects placed in two adjacent corners (20 cm from the walls), and the time spent freely exploring each object was recorded. In the retention trial, one of the two familiar objects was replaced by a novel object, and the time exploring each was recorded. Exploration time (computed when the snout pointed to the object at a distance ≤1 cm) was scored by The Observer XT software (version 11, Noldus, Wageningen, The Netherlands). The discrimination index was calculated using the following formula: (tn − tf)/(tn + tf), where tn represents the amount of time the rats explored the novel object, and tf represents the amount of time rats explored the familiar object.

#### 2.6.8. Morris Water Maze in Young Adults (MWM, PND 68–72)

To assess spatial learning and memory, the rats were tested in the MWM test consisting of two phases: a spatial learning phase (training) of four days duration followed by a single Probe trial 24 h after the training, see Appendix A for a detailed description.

### 2.7. Molecular Analysis

Hippocampi from animals of both sexes not subjected to behavioral tests were collected at PND 23 (Se Opt/Veh = 7 M, 7 F; Se Opt/Pb = 7 M, 7 F; Se Subopt/Veh = 5 M, 6 F; Se Subopt/Pb = 6 M, 6 F) and 70 (Se Opt/Veh = 8 M, 5 F; Se Opt/Pb = 7 M, 5 F; Se Subopt/Veh = 6 M, 6 F; Se Subopt/Pb = 6 M, 6 F) and stored at −80 °C until processing.

#### 2.7.1. Hippocampal Tissues Processing

The total homogenate (HOMO) and the post-synaptic fraction (Triton-insoluble fraction, TIF) of the hippocampal tissues were prepared as previously described in [31]. Briefly, the homogenization of the hippocampi was performed with a Teflon–glass potter in ice-cold lysis buffer (sucrose 0.32 M, Hepes 1 mM, MgCl_2_ 1 mM, NaHCO_3_ 1 mM and PMSF 0.1 mM, at pH 7.4) added to a cocktail of proteases (Roche Diagnostics, Basel, Switzerland) and phosphatases inhibitors (Sigma-Aldrich, Merck KGaA, Darmstadt, Germany). Immediately, an aliquot of HOMO was frozen in dry ice. The remaining HOMO volume followed further steps to obtain the TIF. It was centrifuged at 13,000 *g* for 15 min, the resulting supernatant was removed, and the pellet resuspended in ice-cold buffer (75 mM KCl, 1% Triton-X 100) and finally centrifuged at 100,000× *g* for 1 h. The supernatant was discarded, and the pellet was homogenized in a glass-glass potter in ice-cold Hepes (20 mM) and then stored at −80 °C.

#### 2.7.2. Western Blotting

The protein content of the samples was quantified with a Bio-Rad dye reagent protein assay (Hercules, CA, USA). HOMO and TIF samples were diluted, added to the sample buffer (62.5 mM Tris/HCl pH 6.8, 2% SDS, 10% glycerol, 5% 2-mercaptoethanol, bromophenol blue), and boiled 10 min at 95 °C to obtain protein denaturation. A total of 15 μg of protein per sample were gel-loaded (7.5% Acrylamide/bis-Acrylamide). After gel electrophoresis, proteins were blotted to a nitrocellulose membrane (Bio-Rad, Hercules, CA, USA) and blocked with I-block (Applied Biosystems, Foster City, CA, USA) for 2 h at room temperature. A Western blot analysis was performed after overnight incubation with the following primary antibodies: GluN2A (Sigma Aldrich, Merck KGaA, Darmstadt, Germany), GluN2B (NeuroMab, Davis, CA, USA), GluN1 (Invitrogen, Carlsbad, CA, USA), GluA1 (Calbiochem, Merck KgaA, Darmstadt, Germany), GluA2 (NeuroMab, Davis, CA, USA), PSD-95 (Cell Signaling Technology, Danvers, MA, USA), LRP8 (Abcam, Cambridge, UK), SEPP1 (Abcam, Cambridge, UK), and actin (Sigma Aldrich, Merck KgaA, Darmstadt, Germany). All the primary antibodies were diluted 1:1000, with only the exception of actin (1:2500). Secondary anti-mouse antibody from Sigma Aldrich (Merck KgaA, Darmstadt, Germany) and anti-rabbit antibody from Bio-Rad (Hercules, CA, USA) were diluted 1:10,000 in I-block. Quantification of the Western blotting analysis was performed with Image Lab Software (Bio-Rad, Hercules, CA, USA) after normalization on actin levels. Samples were excluded for technical issues when the signal was too faint to be detected (unmeasurable), and the sample amount did not allow us to repeat the measurement. Groups where samples were excluded are indicated in the respective figure legends. All the reagents, if not differently specified, were purchased from Sigma-Aldrich (Merck KGaA, Darmstadt, Germany).

### 2.8. Statistical Analysis

Behavioral data were analyzed by analysis of variance (ANOVA) with diet, Pb exposure, and sex as between-factors and day/trial/quadrant as repeated measures followed by a post hoc Tukey’s analysis on significant interaction effects. When parametric assumptions were not fully met, the non-parametric Mann–Whitney U-test was used.

Biomarker data were reported as medians and interquartile ranges (IQR: 75th–25th centile). Comparisons between Se levels and Pb exposure (presence/absence) were made by the Mann–Whitney Wilcoxon test. To compare more than two groups, the Kruskal–Wallis test was applied, followed by a Dunn’s post hoc test for multiple comparisons. A *p* value <0.05 (two-tailed) was considered statistically significant.

The statistical analysis of protein expression data was performed within each sex by using an unpaired Student’s *t*-test between Veh and Pb groups or between optimal and suboptimal Se diet groups. Values that deviate from the mean by more than 2.5 standard deviations (SD) were considered outliers and discarded. A 95% confidence interval (*p* < 0.05) was considered statistically significant.

Analyses were performed by SPSS v.27 (IBM Corp., Armonk, NY, USA).

## 3. Results

### 3.1. Reproductive Performances

The data collected at birth did not evidence a main or interaction effect of the Se intake levels on any of the parameters in the study, but they only show a main effect of Pb exposure, irrespective of the Se dietary intake, on litter size: F (1, 30) = 8.303 *p* = 0.0072; Mean ± SD,;Veh = 15.294 ± 2.339; and Pb = 12.529 ± 3.105. However, this effect did not interfere with the pups’ viability, body weight at birth, and sex ratio (see Appendix A for more details).

### 3.2. Plasma and Milk Se Levels

Plasma Se, a reliable biomarker of Se intake when selenomethionine is the dietary Se form, was investigated in dams and offspring (Table 1). Results in dams show that maternal selenium plasma levels pre-mating did not differ significantly between the Se Subopt (Se Subopt/Veh = 6, Se Subopt/Pb = 6) vs. Se Opt (Se Opt/Veh = 5, Se Opt/Pb = 5) groups, which means that a steady state was not achieved for selenium at that stage, notwithstanding acclimation to the relevant Se diet (Subopt or Opt). However, Se plasma levels post-weaning were markedly lower in the two Subopt groups (Se Subopt/Veh = 12, Se Subopt/Pb = 9) compared with the two Opt groups (Se Opt/Veh = 5, Se Opt/Pb = 4). To investigate offspring Se nutrition at delivery, the selenium content of early maternal milk (PND 1) was determined (Table 2). Data show that the offspring of the Subopt groups experienced a poor Se nutrition at birth (Se Opt/Veh = 5, Se Opt/Pb = 4, Se Subopt/Veh = 14, Se Subopt/Pb = 10). The poorer selenium supply in the offspring of the Subopt groups during lactation and after weaning resulted in the lower selenium plasma levels detected up to PND 23 (Se Opt/Veh = 4, Se Opt/Pb = 4, Se Subopt/Veh = 7, Se Subopt/Pb = 9) and PND 70 (Se Opt/Veh = 12, Se Opt/Pb = 4, Subopt/Veh = 17, Se Subopt/Pb = 11), respectively (Table 1).

An influence of Pb exposure on Se plasma levels was detected, with a trend towards higher Se plasma levels in Pb-exposed groups; the difference attained significance only in Se Subopt pre-mating dams and in Se Opt offspring at PND 23 (Table 1).

### 3.3. Blood Pb Levels

Pb blood levels in exposed dams are one order of magnitude higher than in non-exposed ones (Se Opt/Veh = 4, Se Opt/Pb = 4, Subopt/Veh = 16, Se Subopt/Pb = 13, Table 3). In offspring at PND 23, Pb blood levels are twice as much and two orders of magnitude higher that in non-exposed rats (Table 3). At PND 70, long after ceasing Pb exposure (at PND 23), the Pb blood levels of offspring decreased markedly but were not yet at baseline (Se Opt/Veh = 10, Se Opt/Pb = 10, Subopt/Veh = 4, Se Subopt/Pb = 7).

No significant effects of Se nutrition on this biomarker were seen in the exposed groups; however, it is noted that the Subopt group showed markedly less variable Pb blood levels, as indicated by the magnitude of the IQRs (Table 3).

### 3.4. Behavioral Effects of Life-Long Suboptimal Selenium Opmental Lead Exposure in the Neonatal, Juvenile and Adult Offspring

Suboptimal Se nutrition, Pb exposure or their interactive effects were observed in several behavioral domains, except for working memory (measured by the percentage of spontaneous alternations in Y-maze) and spatial learning and memory in the MWM (measured by latency to reach the hidden platform across training days and time spent in the quadrant where the platform was located, respectively). MWM data are reported in Appendix A.

#### 3.4.1. Somatic Growth and Developmental Milestones in Rat Pups

The somatic growth of pups was transiently affected (PND 10–12) only by the combination of suboptimal Se and Pb exposure. Specifically, Se Subopt/Pb pups had higher body weights, diet × treatment × PND interaction: F (3, 204) = 3.638 *p* = 0.0137, and greater body lengths, diet × treatment × PND interaction: F (3, 204) = 6.576 *p* = 0.0003, compared to Se Subopt/Veh pups at PND 10 and 12 (*p* < 0.01 after post hoc comparisons, Appendix A) and did not differ from Se/Opt/Pb group. The analysis performed on body weight at weaning (PND 23) failed to reveal any differences between groups (Appendix A).

As for sensorimotor development, the analysis revealed a significant effect of suboptimal Se on negative geotaxis, F (1, 68) = 23.225 *p* < 0.0001, Figure 2A; Se Subopt pups took more time in orienting in a head-up position compared to Se Opt pups, especially at PND 7, diet × PND interaction F (3, 204) = 5.581 *p* = 0.0011, *p* < 0.01 after post hoc comparisons. The righting reflex was affected by Pb only in female pups that showed longer latency than males to righting on a surface at PND 4, treatment × sex × PND interaction: F (3, 204) = 3.621 *p* = 0.0140, *p* < 0.01 after post-hoc comparisons, Appendix A.

As for spontaneous motor patterns, suboptimal Se, per se, increased the duration of pivoting, diet × PND interaction: F (3, 204) = 2.827 *p* = 0.0397, and wall climbing, main effect of diet: F (1, 68) = 10.907 *p* = 0.0015, diet × PND interaction: F (3, 204) = 4.509 *p* = 0.0044, (Figure 2B,C). Specifically, a developmental delay in the motor abilities of Se Subopt pups may be suggested, as they did not show the expected decrease of pivoting, an immature motor pattern, but exhibited a duration of pivoting at PND 10 comparable to that at PND 4 and 7 (data not shown). Moreover, the interactive effects of suboptimal Se and Pb exposure were observed in locomotion, diet × Pb exposure interaction: F (1, 68) = 6.128 *p* = 0.0158, and head rising, diet × Pb exposure interaction: F (1, 68) = 3.755 *p* = 0.0568; Se Subop/Veh pup rats spent more time in quadrupedal locomotion and head rising compared to either Se Opt/Veh or Se Subopt/Pb pup rats (*p* < 0.01 after post hoc comparisons), Figure 2D,E. Notably, Pb exposure seems to revert the hyperactive profile of Se Subopt rats. The number of USVs emitted by isolated pups was altered by suboptimal Se per se; this is in agreement with the hyperactive profile described above, it increased the number of calls/min emitted [main effect of diet: F (1, 68) = 4.075 *p* = 0.0475, Figure 2F], though all pups showed a similar temporal profile of emissions across PND (Figure 2F).

In the Homing test, the Se Subopt pups reached the arm containing the nest-derived shavings faster than the Se Opt pups (*p* = 0.02). However, they spent the same amount time in the nest arm compared to Se Opt pups (Appendix A).

#### 3.4.2. Locomotor and Explorative Activity in Juvenile Rats

The interactive effects of Suboptimal Se and Pb exposure were found in the OF test. Specifically, as for locomotor activity, Se Subopt/Pb rats travelled shorter distances, diet × treatment interaction: F (1, 72) = 4.16 *p* = 0.04, and moved with lower mean velocities, diet × treatment interaction: F (1, 72) = 4.65 *p* = 0.03, compared to Se Subopt/Veh and Se Opt/Pb rats (*p* < 0.01 after post hoc comparisons, Figure 3A,B).

As for anxiety-like behavior, Se Subopt/Pb rats spent less time in the central zone compared to Se Subopt/Veh, diet × Pb interaction: F (1, 72) = 5.207 *p* = 0.02, *p* < 0.01 after post hoc comparisons, Figure 3C.

Suboptimal Se per se, Pb exposure, or their interaction did not affect spontaneous alternation or the total number of arm entries in Y-maze (Figure 3D,E). However, the Se suboptimal diet per se exerted a significant activating effect on ethological measures describing exploration and arousal levels, such as open rearing, frequency: F (1, 72) = 6.179 *p* = 0.01; duration: F (1, 72) = 14.554 *p* = 0.0003, wall rearing, frequency: F (1, 72) = 3.902 *p* = 0.05; duration: F (1, 72) = 30.647 *p* < 0.0001, and grooming, frequency: F (1, 72) = 11.926 *p* = 0.0009; duration: F (1, 72) = 16.400 *p* < 0.0001, with a general increase of all these behaviors in Se Subopt rats compared to Se Opt rats (Figure 3F–H).

#### 3.4.3. Anxiety-Like Behavior and Locomotor Activity in Adult Rats

Interactive effects of suboptimal Se and Pb exposure on anxiety-like behavior detected at the juvenile stage were also observed in adulthood. Se Subopt/Pb rats spent less time in the open arms of the EPM compared to Se Subopt/Veh rats, diet × Pb exposure interaction: F (1, 72) = 2.969 *p* = 0.08, *p* < 0.05 after post hoc comparisons, Figure 4A.

The effects of Se nutrition and Pb combination on anxiety-like behavior was also evident in the OF test (NOR habituation): Se Subopt/Pb rats spent less time in the center of the arena compared to Se Opt/Pb rats, diet × Pb exposure interaction: F (1, 72) = 5.145 *p* = 0.02, *p* < 0.05 after post hoc comparisons, Figure 4B.

Suboptimal Se, regardless of Pb exposure, caused a decrease in locomotor activity at the adult stage, which was evident only after Pb exposure at the juvenile stage, distance: main effect of diet F (1, 72) = 5.539 p = 0.02; mean speed: main effect of diet F (1, 72) = 5.942 *p* = 0.01, (Figure 4C,D).

#### 3.4.4. Long-Term Recognition Memory in Adult Rats

Suboptimal Se did not affect long-term recognition memory in NOR. However, we found a significant Pb effect in NOR but in female sex only, treatment × sex interaction: F (1, 72) = 9.52 *p* = 0.0029. Female rats exposed to Pb failed to show a preference for the novel object, irrespective of the Se diet received, exhibiting a lower discrimination index compared to either Veh female rats or Pb male rats (*p* < 0.01 after post hoc comparisons), (Figure 4E).

### 3.5. Impact of Suboptimal Selenium Intake on the Expression of NMDA and AMPA Receptors Subunits, LRP8 and SEPP1, in the Hippocampus of Juvenile and Adult Offspring

#### 3.5.1. Expression and Trafficking to the Post-Synaptic Site of the NMDA and AMPA Receptor Subunits in the Hippocampi of Juvenile Rats: Optimal vs. Suboptimal Se

The protein expression of NMDAR and AMPAR subunits of the glutamatergic receptors was evaluated in the hippocampus of PND 23 rats born from dams fed with the optimal or suboptimal Se diet.

Males and females were analyzed separately, and protein expression was revealed by Western blot, both in the total homogenate (HOMO) and Triton Insoluble Fraction (TIF), representative of the post synapses. When gender differences were not observed, male and female groups were combined.

Our analyses at PND 23 revealed a general reduced expression of constitutive elements of NMDAR, GluN2A, and GluN2B subunits, as well as AMPAR, GluA1, and GluA2 (Figure 5). The effect was significant in the total homogenate of Se Subopt/Veh (Figure 5A,B). Modifications in the total homogenate were reflected, with minor differences, at the post-synaptic fraction of the PND 23 hippocampi (Figure 5D,E). The diet did not affect the GluN2B/GluN2A ratio, both in HOMO and TIF (HOMO: mean ± SEM were 0.81 ± 0.10 for Opt Se (*n* = 11) versus 0.79 ± 0.16 for Subopt Se (*n* = 9); TIF: mean ± SEM were 2.70 ± 0.44 for Opt Se (*n* = 12) versus 3.70 ± 0.5 for Subopt Se (*n* = 9); unpaired *t*-test not significant).

Contrary to what was observed for GluN2A and GluN2B, GluN1 subunit levels are similar in both diets but significantly increase at the post-synapse in PND 23 hippocampi (Figure 5D).

The decrease in the expression of several subunits of the NMDAR and AMPAR was also coupled with a reduced expression of PSD-95 (Figure 5C).

#### 3.5.2. Expression and Trafficking to the Post Synaptic Site of NMDA and AMPA Receptor Subunits in the Hippocampi of Adult Rats: Optimal vs. Suboptimal Se

At PND 70, our results show a recovery in Se Subopt/Veh animals for most subunits that returned to the levels of optimal Se in HOMO. A significant increase for GluN2A and GluA1 was observed in the TIF of Se Subopt/Veh animals compared to Se Opt/Veh (Appendix A). No sex differences were observed for the NMDA and AMPAR subunits in Appendix A, except for the GluN2B and GluA2 subunits.

GluN2B and GluA2 expression was significantly increased in HOMO of Se Subopt/Veh female rats compared to Se Opt/Veh (Appendix A). A similar tendency is evident in males, although it is not significant. A significant increase of GluA2 occurred instead in the TIF of Se Subopt/Veh male rats compared to Se Opt/Veh (Appendix A).

Different from what was observed at PND 23, PSD-95 increased at PND 70 in Se Subopt/Veh compared to Se Opt/Veh rats in both sexes (Appendix A).

#### 3.5.3. Expression of SEPP1 and Expression and Trafficking to the Post Synaptic Site of LRP8 in the Hippocampus: Optimal vs. Suboptimal Se

Se transport in the brain and delivery to neurons are mainly under the control of SEPP1 and LRP8 (for an extensive review see [9]). To evaluate whether these two functions might have been compromised by a prolonged suboptimal Se diet, we analyzed SEPP1 and LRP8 expression in the hippocampi of 23 and 70 PND rats of both sexes. LRP8 levels were monitored both in HOMO and TIF. SEPP1 was only monitored in total homogenate, this being a soluble protein.

No gender differences were observed both in LRP8 and SEPP1 expression. The different supply of Se in the optimal vs. suboptimal diets did not affect the total expression (HOMO) of SEPP1 and LRP8 in the hippocampus of 23 PND rats (Figure 6A,B). Nevertheless, LRP8 was significantly decreased in the TIF obtained from hippocampi of Subopt Se rats at PND 23 (Figure 6C). Although a Se diet affects the level of LRP8 similarly in males and females, a significantly higher amount of LRP8 is evident in females compared to males at a Se optimal diet. Any difference in LRP8 due to the diet or sex evident at PND 23 disappeared in adulthood (Appendix A).

### 3.6. Impact of Selenium Intake on the Effect of Lead on NMDA and AMPA Receptors and Se Transporters in the Hippocampus of Juvenile and Adult Offspring

To evaluate whether Se intake might modulate Pb effect on NMDA and AMPAR subunits, we analyzed the hippocampi of male and female rats fed with the optimal or suboptimal Se diet and exposed to Pb in drinking water at a dose of ~1.25 mg/kg bw Pb per day. Since we previously demonstrated that developmental exposure to Pb in drinking water at 5 mg/kg bw per day, in the presence of optimal Se nutrition, induces sex dimorphic effects [29], males and females were analyzed separately.

#### 3.6.1. Expression and Trafficking to the Post-Synaptic Site of NMDA and AMPA Receptors Subunits in the Hippocampi of Rats Fed with Optimal Se and Exposed to Pb

The effect of Pb on the expressions of the NMDAR and AMPAR subunits in HOMO and levels in TIF was initially evaluated in animals with optimal Se. No effect was evident in either male and female total homogenates for all the subunits tested at PND 23 (Appendix A).

A Pb effect emerged analyzing the post-synaptic site, and it was found to be sex dependent. In particular, the GluN2A and GluN2B subunits of NMDAR (Figure 7A), as well as the GluA1 and GluA2 subunit levels of AMPAR (Figure 7B), increased the TIF of Pb-exposed females only. In males’ TIF, these subunits remained unaffected (Figure 7C,D), but we observed a modulation of GluN1, which significantly increases upon exposure to ~1.25 mg/kg bw Pb per day (Figure 7C).

No Pb-induced modifications of PSD-95 expression were observed, regardless of sex, at PND 23 (Appendix A). Moreover, Pb did not affect any of the analyzed subunits at PND 70 in both sexes (Appendix A).

#### 3.6.2. Expression and Trafficking to the Post-Synaptic Site of NMDA and AMPA Receptors Subunits in the Hippocampi of Rats Fed with Suboptimal Se Diet and Exposed to Pb

The interactive effects of suboptimal Se and Pb exposure on the hippocampal glutamatergic receptors were only significant in males at PND 23. GluA2 was the only subunit significantly increased in HOMO after exposure to ~1.25 mg/kg bw Pb per day (Figure 8B). Despite the increased expression of GluA2, a significant decrease is evident in TIF (Figure 8E). Such a decrease at the post-synaptic site is coupled with a decrease in GluA1 (Figure 8E) and a significant increase in GluN2B and GluN1 (Figure 8D), which were not altered in HOMO (Figure 8A), and PSD 95 (Figure 8C) at PND 23.

Different from males, Pb did not affect either the total expression or the spine distribution of the NMDAR and AMPAR subunits, nor did it affect the PSD 95 amount in female rats fed the suboptimal Se diet at PND 23 (Appendix A).

In addition to a persistent downregulation of GluA1 and GluA2 subunits in hippocampal TIF of male rats fed with suboptimal Se, no further effect was evident at PND 70 in both males and females (Appendix A).

## 4. Discussion

In this study, we established a rat model of Se restriction and developmental exposure to a low Pb dose, to disentangle the effects of suboptimal Se nutrition and developmental Pb exposure on behavior and hippocampal neuroplasticity. Biomarker data (i.e., plasma and milk Se, blood Pb) confirmed the validity of our model to achieve suboptimal Se status and low-level Pb exposure.

First of all, our study indicates that different behavioral domains are targeted by Se in the developing organism and provides evidence of the potential interaction of the Se status with an adverse environmental challenge affecting the developing brain, represented by Pb exposure.

Se suboptimal nutrition, per se, induces alterations in sensorimotor and emotional development. Notably, some of the motor behavioral alterations observed during the first weeks of life (i.e., increased wall climbing) may be also considered to be early indicators of the long-term repetitive/stereotypic-like behaviors found in juvenile/adult rats. Indeed, a substantial increase in vertical explorative activity and self-grooming was evident in juveniles in the Y-maze test. Similarly, USVs emitted by isolated pups may be interpreted as distress, the expression of an early anxiety state, and increased emotional arousal [32,33].

At the molecular level, suboptimal Se nutrition, per se, induces the downregulation of the NMDAR and AMPAR subunit expressions, except for GluN1, which is mirrored at the post-synapse and coupled to a reduction of PSD-95. Most of the effects are evident in the juvenile hippocampus and partially reverted at PND 70.

During neuronal development, GluN2 subunits’ compositions vary in the hippocampus, leading to a predominance of GluN2A over GluN2B in mature neurons [34]. The switch between GluN2B and GluN2A, resulting in a decrease in the ratio GluN2B/GluN2A, is part of the program of synapse maturation, circuit refinement, and the acquisition of learning abilities [34]. Despite the significant reduction of GluN2A and GluN2B subunits, the suboptimal intake of Se does not affect this ratio that remains similar to that observed at optimal Se. Still, the observed general unbalance is probably reflected in a reduced amount of PSD95. PSD-95 is another key player in synapse maturation and stabilization driving trafficking and localization of glutamate receptors and favoring clusterization with different signaling molecules [35]. An increase in PSD-95 accompanies spine enlargement upon LTP and is required for the activity-dependent stabilization of excitatory synapses [36,37]. In addition, the increased expression of the GluN1 subunits observed only at 23 PND may suggest the prevalence in sub-optimal conditions of heterodimers different from the classical GluN1/GluN2B and GluN1/GluN2A. As a mandatory subunit of the NMDAR, a GluN1 subunit may form heterodimers with GluN2C-D or GluN3A-B. NMDAR subunits express differently along development and distribute in different brain areas [38]. In particular, GluN2D and GluN3A are expressed in early development [38] and have been detected in the hippocampus [39].

Our results thus suggest that the general decrease of glutamatergic subunits involved in plasticity and the possible unbalance in heterodimers composition, due to the Se shortage, might counteract the stabilization of dendritic spines, delaying the maturation of the excitatory network. This effect is reverted in adulthood. Indeed, PSD-95 expression significantly increases at PND 70 in both males and females. At PND 70, the expression of GluN2A, GluN1, and AMPAR subunits is similar to that observed at optimal Se levels in both sexes.

An important feature of the CNS is to be protected from Se deficiency [40]. As such, we observed a tendency, although not significant, to increase total LRP8 expression in Se Subopt/Veh compared to Se Opt/Veh rats. Still, we found a drop in LRP8 in TIF. This finding raises a question about the specific role that Se could play in the post-synaptic compartment. LRP8 is not only the main player in the entrance of Se into the central nervous system through the BBB, but also plays a role in the intra-neuronal uptake of SEPP1, the extracellular circulating selenoprotein which transports Se across the whole body [9]. The peculiarity of the LRP8 function at TIF should be examined in depth in further studies. Our data match with recent findings by Zhang and coworkers (2021) which demonstrated that selenomethionine treatment prevents hippocampal synaptic loss, ameliorating synaptic deficits through mechanisms that linked selenoproteins, the glutamatergic machinery, particularly NMDARs, and the calcium ion influx [41].

Interestingly, a suboptimal Se status can unmask the effects of a low Pb exposure on selected behavioral parameters; for example, the depressant effects of Pb on locomotor activity in pups and juveniles are evident only in the condition of suboptimal Se. Moreover, persistent anxiety-like alterations (assessed in both OF and EPM through the reluctance to enter into areas perceived as unsafe) were evident only in Pb offspring fed suboptimal Se.

Whereas a suboptimal Se status per se was not sufficient to outline sex dimorphic effects, sex-related differences emerged when we considered Pb exposure.

At the behavioral level, Pb, regardless of Se status, affected the long-term recognition memory differently in the two sexes, impairing novel object discrimination in the female animals only. This finding is consistent with our previous study on a rat model of developmental exposure to Pb at 5 mg/kg bw per day [29]. Likewise, only female rats with optimal Se that were exposed to Pb showed increased translocation of NMDAR and AMPAR subunits to the TIF. The altered levels of specific subunits in hippocampal TIF, in the absence of an altered total expression (i.e., in HOMO), suggests that, despite an adequate intake of Se, Pb exerted an effect in female pups through the finely modulated alteration of receptor trafficking inside and outside the spine. All the Pb-induced effects in female rats fed with optimal Se observed at PND 23 were completely abolished in the long term (PND 70), and no other modulation occurred. In males, Pb induces only an increase in GluN1 distribution to the post-synaptic site at PND 23, with no other alterations occurring in adulthood.

Surprisingly, females with suboptimal Se that were exposed to Pb did not show any alterations in glutamatergic machinery both at PND 23 and 70. On the contrary, males appeared vulnerable to the combination of a Se suboptimal diet and developmental Pb exposure. Specifically, in males, we observe the upregulation of GluA2 expression in the cellular homogenate with major effects in TIF where there is an increase in GluN2B and GluN1 translocation followed by a decrease in GluA1 and GluA2 at the post-synaptic site (PND 23). The drop in AMPAR synaptic translocation is consistent with the increase of post-synaptic GluN2B. In fact, along with physiological development, a decrease in the GluN2B amount at the post-synapse is critical for proper synapse maturation and AMPAR insertion at the post-synapse [42]. The decreased translocation of GluA1 and GluA2 remains impaired at PND 70, suggesting that the interactive effects of suboptimal Se and Pb exposure last in the long term. The lack of Pb effects in males at optimal Se intake suggests that Se exerts a sex-dependent protective effect against Pb exposure.

Overall, these data indicate that a suboptimal Se intake induces a generalized drop in the expression of the glutamatergic receptors without sex differences; Pb instead induced alterations in the amount of NMDAR and AMPAR in males (suboptimal Se) and females (optimal Se), acting on different receptor subunits with a sexually dimorphic effect dependent on the Se supply. The Pb effect specifically occurs at the post-synaptic site, suggesting a fine mechanism that could impact spine maturation and, consequently, hippocampal neuroplasticity.

Our biomarker data suggest that Pb exposure influenced Se plasma levels, since higher Se plasma levels were apparent in Pb-exposed groups. Indeed, SEPP1 is known to form complexes with heavy metal ions, and our data would support an increased expression of SEPP1 in the liver in response to high circulating Pb levels. Recent evidence on the distribution of Pb and Se in mouse brains following sub-chronic Pb exposure shows that the two trace elements may interact in vivo [43]. Pb was found to be mainly deposited as particles in brain slices, and, in the Pb-exposed animals, Se correlated significantly with Pb in these particles in the cortex and hippocampus/corpus callosum regions, with a molar ratio close to 1:1. These results were interpreted as an indication that Se may play a crucial role in Pb-induced neurotoxicity.

Taking stock of these considerations, the hypothesis can be put forward that Se interacts with Pb chemically, which may result in detoxification but at the expense of Se-depletion, which may in turn negatively affect neurodevelopment [43]. As a whole, and in line with epidemiological findings, our data indicate that a suboptimal Se status during pregnancy adversely influences motor competencies and emotionality in early life and modulates the effects of Pb on somatic growth in pre-weaned pups and on locomotor activity and anxiety at later life stages. It is worth noting that several human studies have demonstrated the involvement of a Se deficiency in mood disorders [43,44]. Our present findings are also in line with previous experimental studies showing significant changes in locomotor activity and anxiety-like behavior in animals fed on a Se-deficient diet [44,45], although it is remarkable that, in the present study, the findings were seen as a consequence of a suboptimal Se nutrition and not of an overt Se deficiency.

## 5. Conclusions

The results of our research point out the possible outcomes of a suboptimal Se supply during critical brain-developmental windows. The present study shows that Se suboptimal nutrition, apart from directly affecting neurobehavioral development and altering the molecular profile of glutamatergic receptors, may also interact with Pb exposure and worsen Pb’s adverse effects on the developing brain and behavior. Whether this may happen via mechanisms, possibly entailing a direct chemical interaction, is an open research question.

## Figures and Tables

**Figure 1 nutrients-14-03269-f001:**
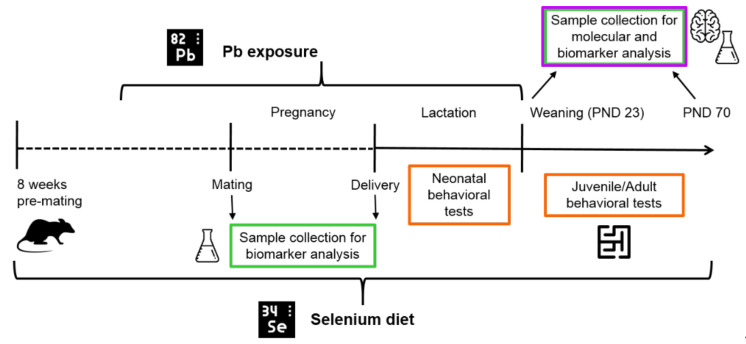
Timeline of experimental design. Female rats were fed an optimal (0.15 mg/kg) or suboptimal (0.04 mg/kg) Se diet for 8 weeks pre-mating, and, after 4 weeks, they were exposed until the end of lactation to 12.5 µg/µL Pb via drinking water. At weaning (post-natal day, PND 23), offspring were fed on the same diet as their respective dams until adulthood and completion of the behavioral assessment. Behavioral testing (neonatal and juvenile/adult tests) and tissue collection for Se and Pb determinations and molecular analysis of hippocampus were performed at the time points indicated.

**Figure 2 nutrients-14-03269-f002:**
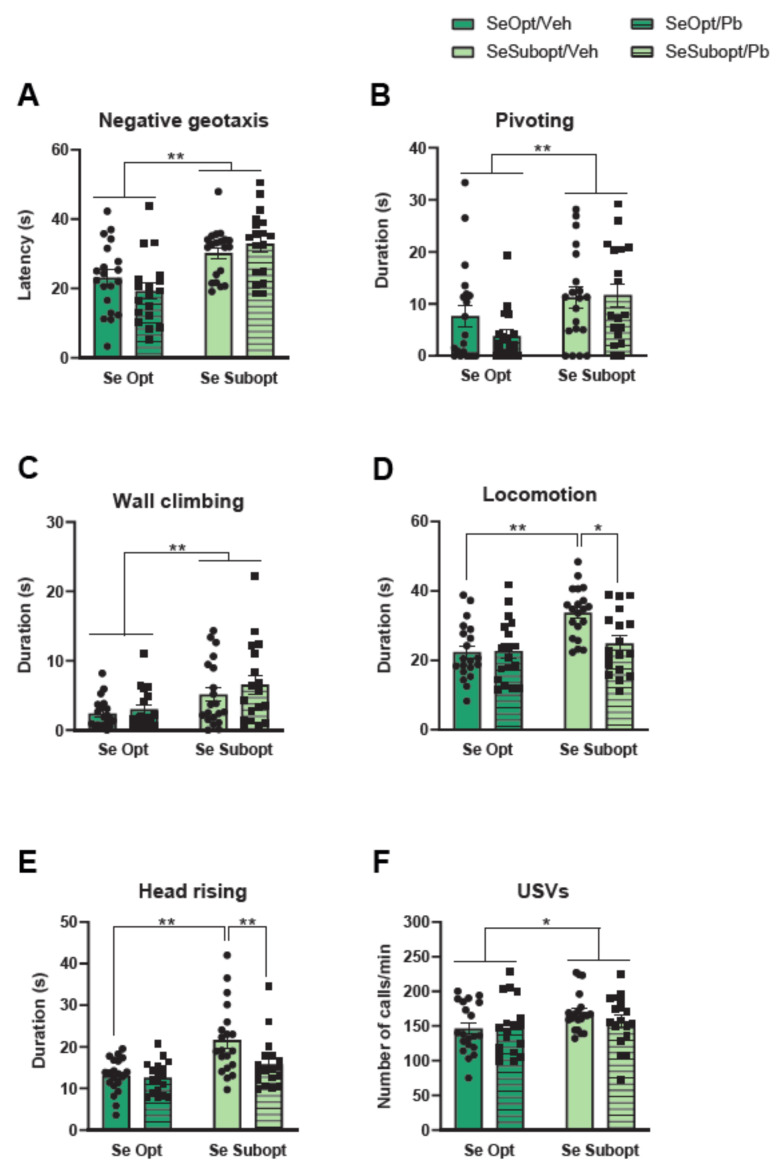
Neonatal assessment in pups at post-natal day (PND) 4, 7, 10, and 12. (**A**) Mean latency to perform negative geotaxis (values pooled across PND 4–12); (**B**) pivoting duration at PND 10; (**C**) total duration of wall climbing, (**D**) locomotion, and (**E**) head rising (values pooled across PND 4–12); (**F**) mean number of USVs emission (values pooled across pnd 4–12). Data are shown as individual value plots and are sex-pooled represented; * *p* < 0.05, ** *p* < 0.01.

**Figure 3 nutrients-14-03269-f003:**
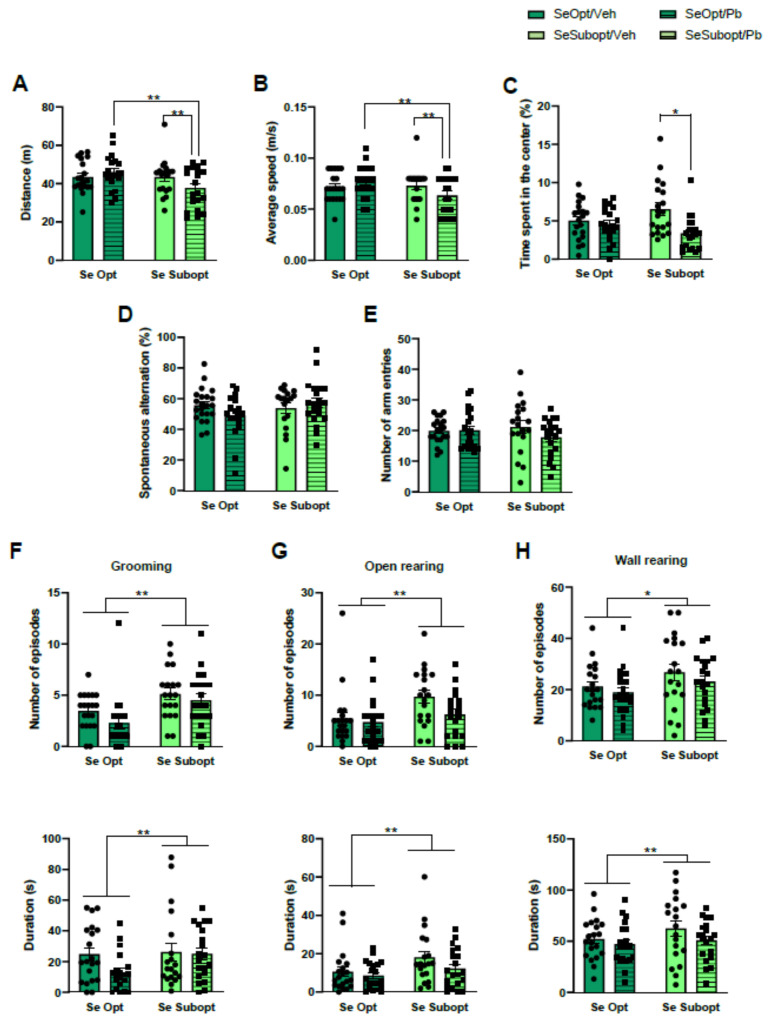
Behavioral assessment in juvenile rats. Locomotor activity measured by (**A**) Distance travelled and (**B**) average speed, and (**C**) anxiety-like behavior measured by time spent in the center during the Open-Field (OF) test; working-memory performance measured by (**D**) percentage of spontaneous alternation, and explorative activity measured by € number of entries, (**F**) open rearing, (**G**) wall rearing, and (**H**) grooming during Y-maze test. Data are shown as individual value plots and are sex-pooled represented; * *p* < 0.05, ** *p* < 0.01.

**Figure 4 nutrients-14-03269-f004:**
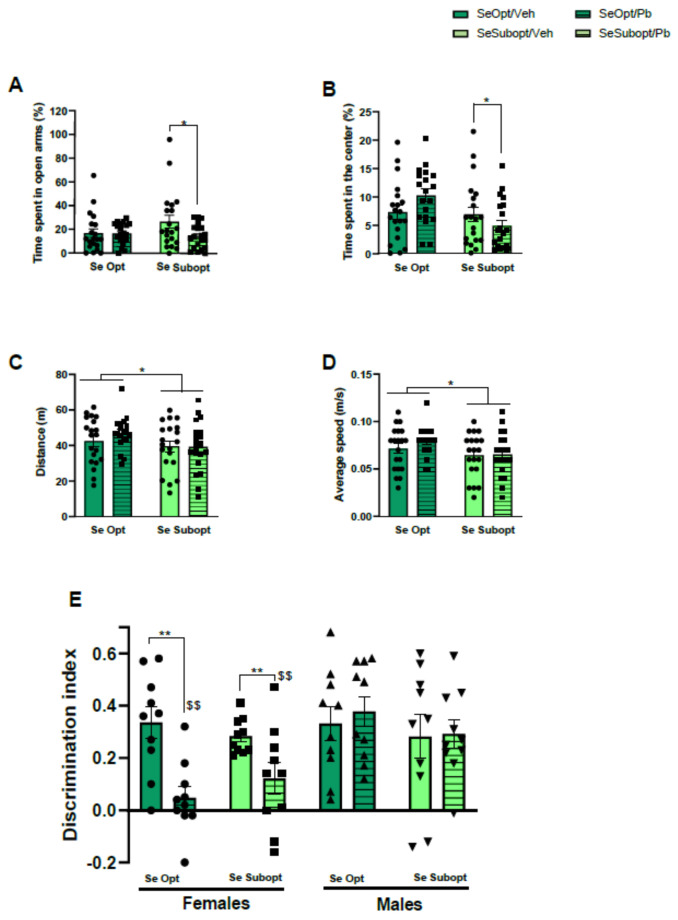
Behavioral assessment in young adult rats. Anxiety-like behavior measured by percentage of time spent in (**A**) open arms during the Elevated Plus Maze (EPM) test and in (**B**) the center of the arena during the Open-Field (OF) test; locomotor activity measured by (**C**) distance travelled and (**D**) average speed in the OF; long-term recognition memory performance measured by (**E**) discrimination index in the Novel Object Recognition (NOR) test. Data are shown as individual value plots and are sex-pooled represented, except for NOR test; * *p* < 0.05, ** *p* < 0.01, $$ *p* < 0.01 Pb females vs. Pb males.

**Figure 5 nutrients-14-03269-f005:**
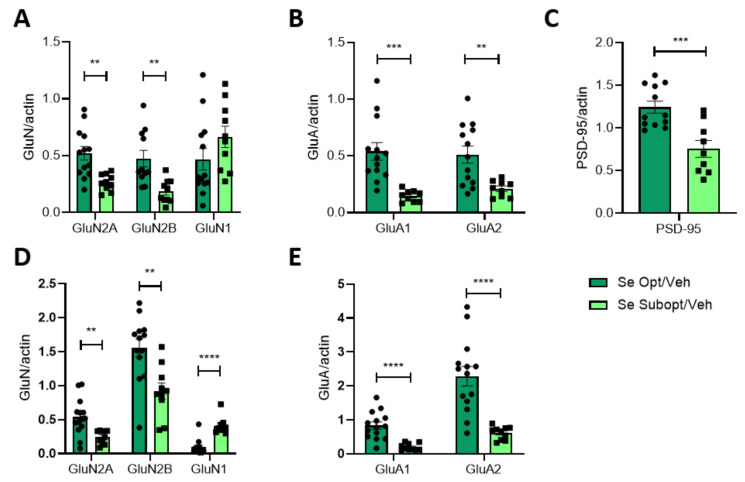
Effect of Se diet on the protein expression and post-synaptic trafficking of glutamatergic receptors and PSD-95 in juvenile rats. Quantification of NMDA and AMPA receptor subunits in the hippocampus of PND 23 rats homogenate (**A**–**C**) and TIF (**D**,**E**). Animals were fed with optimal or suboptimal Se diets. Data are shown as individual value plots and are represented as ratios of actin levels (*n* ≥ 7). Unpaired *t*-test, ** *p* < 0.01, *** *p* < 0.001, **** *p* < 0.0001. Opt-Se: GluN2B *n* = 11, GluN1 *n* = 12 due to technical issues.

**Figure 6 nutrients-14-03269-f006:**
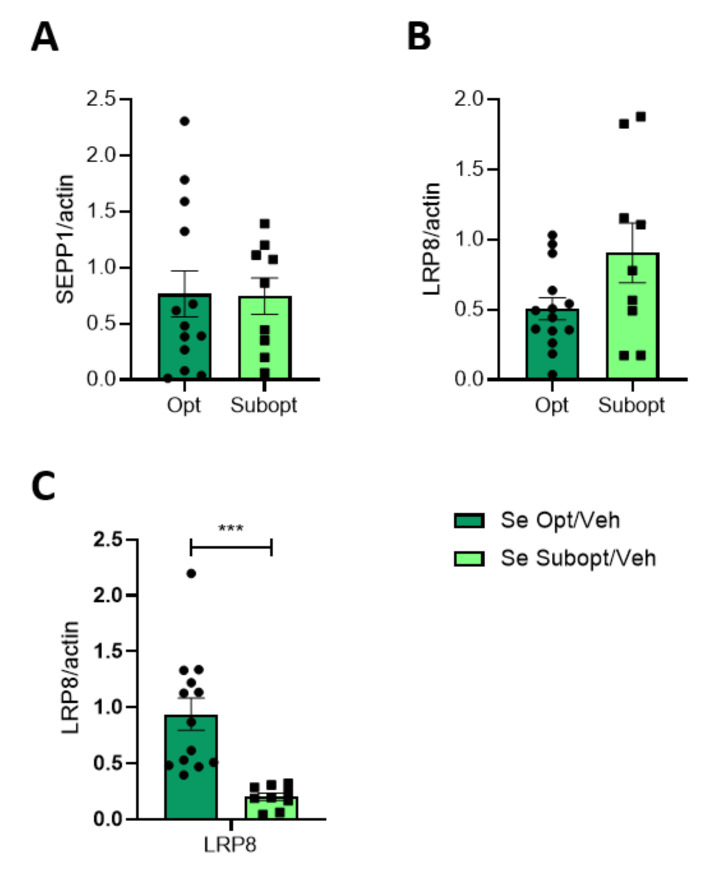
Effect of Se diet on selenoprotein and Se transporter protein expression in juvenile rats. Protein expression of SEPP1 (**A**) and LRP8 (**B**) in the homogenate and TIF (LRP8, (**C**)) of PND 23 hippocampi of rats fed with optimal versus suboptimal Se diet. Data are shown as individual value plots and are represented as ratio on actin levels (*n* ≥ 9); Unpaired *t*-test, *** *p* < 0.001.

**Figure 7 nutrients-14-03269-f007:**
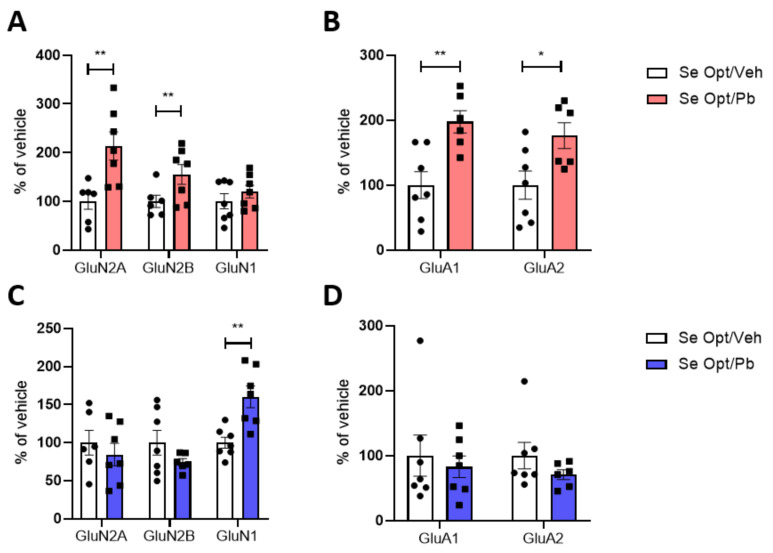
Sex differences in the post-synaptic trafficking of glutamatergic receptors in juvenile rats fed with optimal Se diet and exposed to Pb. Expression of NMDA and AMPA receptors in TIF of PND 23 hippocampi of female (**A**,**B**) and male (**C**,**D**) rats fed with optimal Se diet and exposed to ~1.25 mg/kg body weight Pb per day. Data were normalized on actin levels. Data are shown as individual value plots and are represented as % of vehicle (*n* ≥ 6); Unpaired *t*-test, * *p* < 0.05, ** *p* < 0.01.

**Figure 8 nutrients-14-03269-f008:**
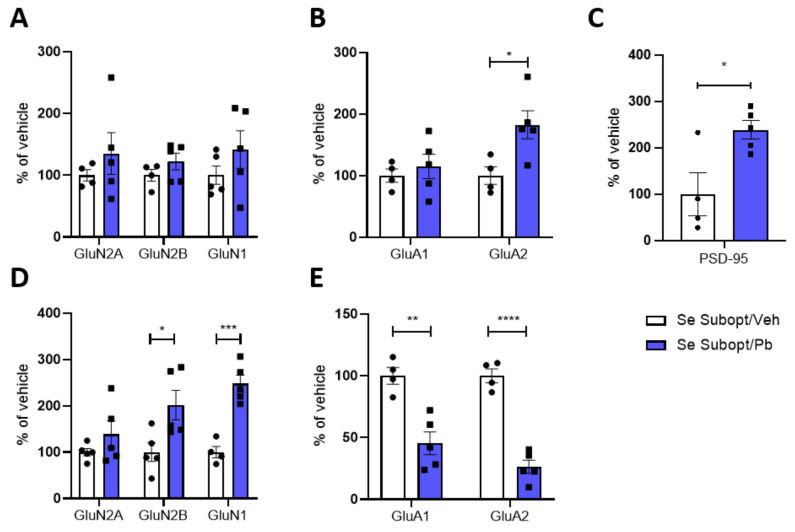
Effect of suboptimal Se diet and Pb exposure on glutamatergic receptors expression and post-synaptic trafficking in male juvenile rats. Expression of NMDA and AMPA receptors and PSD-95 in the homogenate (**A**–**C**) and TIF (**D**,**E**) of PND 23 hippocampi of male rats fed with suboptimal Se diet and exposed to ~1.25 mg/kg body weight Pb per day. Data were normalized on actin levels. Data are shown as individual value plots and are represented as % of vehicle (*n* ≥ 4); Unpaired *t*-test, * *p* < 0.05, ** *p* < 0.01, *** *p* < 0.001, **** *p* < 0.0001.

**Table 1 nutrients-14-03269-t001:** Se concentrations in plasma of dams and offspring (µg Se/mL): medians and IQRs.

Group		Plasma Se Levels			
		Dams		Offspring	
		Pre-Mating	Post-Weaning	PND 23	PND 70
Se Subopt	Veh	0.265 (0.10)	0.167 (0.06)	0.040 (0.02)	0.130 (0.10)
	Pb	0.302 (0.09) °°°	0.147 (0.03)	0.053 (0.01)	0.164 (0.07)
Se Opt	Veh	0.319 (0.04)	0.352 (0.04) ***	0.169 (0.03) ***	0.454 (0.10) ***
	Pb	0.329 (0.04)	0.381 (0.08) **	0.252 (0.03) **°	0.522 (0.17) **

** *p* < 0.01 vs. Se Subopt/Pb. *** *p* < 0.001 vs. Se Subopt/Veh. ° *p* < 0.05 vs. Se Opt/Veh. °°° *p* < 0.001 vs. Se Subopt/Veh.

**Table 2 nutrients-14-03269-t002:** Se concentrations in early maternal milk (µg Se/mL): medians and IQRs.

Group		Milk Se Levels
Se Subopt	Veh	0.046 (0.01)
	Pb	0.047 (0.03)
Se Opt	Veh	0.067 (0.04) ***
	Pb	0.100 (0.03) **

*** *p* < 0.001 vs. Se Subopt/Veh. ** *p* < 0.01 vs. Se Subopt/Pb.

**Table 3 nutrients-14-03269-t003:** Pb concentrations in blood of dams and offspring (µg Pb/mL): medians and IQRs.

Group		Blood Pb Levels		
		Dams	Offspring	
		Post-Weaning	PND 23	PND 70
Se Subopt	Veh	0.002 (0.00)	0.003 (0.00)	0.003 (0.01)
	Pb	0.089 (0.03) ***	0.180 (0.07) ***	0.006 (0.00)
Se Opt	Veh	0.002 (0.00)	0.002 (0.00)	0.001 (0.01)
	Pb	0.065 (0.04) ***	0.168 (0.11) *	0.006 (0.01) **

* *p* < 0.05 vs. Opt/Veh. ** *p* < 0.01 vs. Opt/Veh. *** *p* < 0.001 vs. Subopt/Veh.

## Data Availability

The data presented in this study are available on request from the corresponding author.

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
