# Peer review of "Short- and Long-Term Effects of Suboptimal Selenium Intake and Developmental Lead Exposure on Behavior and Hippocampal Glutamate Receptors in a Rat Model"

_nutrients, 2022, doi:10.3390/nu14163269_

Round 1

Reviewer 1 Report

Please illustrate diagrammatically the mechanism of selenium in brain development. Make your diagram clear, concise, and informative. Remember, not every reader is a scientist. The conclusion must be revised to meet journal standards. What is the exact role of selenium in CNS development? State it clearly in the conclusion. 

Author Response

According to Reviewer #1

Please illustrate diagrammatically the mechanism of selenium in brain development. Make your diagram clear, concise, and informative. Remember, not every reader is a scientist. The conclusion must be revised to meet journal standards. What is the exact role of selenium in CNS development? State it clearly in the conclusion.

Answer: We would like to thank the reviewer for the suggestion. We added a graphical abstract to the manuscript to help all the readers to understand the focus of our work, the role of Se and Pb in neurodevelopment and their effects in the molecular and behavioral domains. We also added little specifications in the conclusions to highlight what is reported in details in the discussion.

Reviewer 2 Report

The authors in this study determined “Short- and long-term effects of suboptimal selenium intake and developmental lead exposure on behavior and hippocampal glutamate receptors in a rat model”. The current study demonstrated the investigation of short- and long-term effects of early life low Se supply on behavior and synaptic plasticity with a focus on hippocampus, considering both suboptimal Se intake per se and its interaction with developmental exposure to lead (Pb). n rat offspring the assessment of motor, emotional and cognitive endpoints at different life stages was complemented by the evaluation of expression and synaptic distribution of NMDA and AMPA receptor subunits at PND 23 and 70 in the hippocampus. Therefore, Suboptimal Se status co-occurring with Pb exposure induced a transient body weight increase and a persistent anxiety-like behavior. From the molecular point of view, we observed hippocampal alterations in NMDA (Glun2B and GluN1) and AMPA receptor subunits trafficking to the post-synapse in male rats only. Our study provides evidence of potential Se interaction with Pb in the developing brain 

In my opinion, there are scarce experimental results generated in this study design regarding the mechanism of neurotoxicity. However, the available research information seems to be insufficient in the current form of the manuscript. Therefore, these results cannot support the mechanism proposed by these authors. Although the authors have shown hippocampal alterations in NMDA (Glun2B and GluN1) and AMPA receptor subunits trafficking to the post-synapse. The observed result in this study is cannot completely exhibit the mechanism of the study. Therefore, there is no direct experiments to support this hypothesis for the mechanism of the study.

1)    In figure 2 and 3 the authors have shown the behavioral results for various experiments, they have used Stoelting software for behavior assessment. The authors need to show the animal tracking pictures in the figure to present their novel finding.

2)    In figure 4 the authors need to show the animal tracking pictures in the figure for open field, elevated plus maze and NOR test.

3)    In figure 5, quantification of NMDA and AMPA receptors, PSD 95 is the assessment of protein or what expression levels?

4)    In Figure 6, quantification of SEPP1 and LRP8 in the homogenate, what expression is that protein or gene, kindly explain?

5)    This manuscript needs serious revision, the authors need to give separate section for abbreviation. Many words are irregular and not first time abbreviated. The abbreviation leads the readers in trouble to understand the results.

6)    The authors can also add this reference for the methods of behavioral experiments, which can merit their present study https://doi.org/10.1016/j.redox.2022.102280, https://doi.org/10.1016/j.apsb.2022.01.017

7)    I request the authors to illustrate a schematic diagram to demonstrate that hippocampal alterations in NMDA (Glun2B and GluN1) and AMPA receptor subunits trafficking to the post-synapse mechanism, with conclusive evidence, for a lay man understanding to the readers.

8)    The authors need to carefully revise the manuscript with the requested experiments and a language check would be better.

Even though the authors describe those hippocampal alterations in NMDA (Glun2B and GluN1) and AMPA receptor subunits trafficking to the post-synapse mechanism with conclusive evidence. There is scarce evidence for the current hypothesis and understanding, the authors can majorly revise the manuscript and resubmit for further review. In current form the manuscript cannot be published in Nutrients.

Author Response

According to Reviewer #2

In my opinion, there are scarce experimental results generated in this study design regarding the mechanism of neurotoxicity. However, the available research information seems to be insufficient in the current form of the manuscript. Therefore, these results cannot support the mechanism proposed by these authors. Although the authors have shown hippocampal alterations in NMDA (Glun2B and GluN1) and AMPA receptor subunits trafficking to the post-synapse. The observed result in this study is cannot completely exhibit the mechanism of the study. Therefore, there is no direct experiments to support this hypothesis for the mechanism of the study.

1)    In figure 2 and 3 the authors have shown the behavioral results for various experiments, they have used Stoelting software for behavior assessment. The authors need to show the animal tracking pictures in the figure to present their novel finding.

Answer: We thank the reviewer for the suggestions. In order to highlight our findings relative to the effects of Se and Pb on locomotor activity, in the revised manuscript we added the animal tracking pictures in the graphical abstract.

2)    In figure 4 the authors need to show the animal tracking pictures in the figure for open field, elevated plus maze and NOR test.

Answer: We analyzed the Open-Field test only using ANY-Maze video-tracking software (Stoelting Europe, Dublin, Ireland). We added the relative pictures in the graphical abstract as stated above.

The other behavioral tests (Elevated Plus Maze and Novel Object Recognition test) were analyzed by a blinded experimenter using The Observer XT software, a behavioral coding software that it does not provide  tracking pictures, but only data concerning frequency, duration and latency of selected behavioral items.

3)    In figure 5, quantification of NMDA and AMPA receptors, PSD 95 is the assessment of protein or what expression levels?

4)    In Figure 6, quantification of SEPP1 and LRP8 in the homogenate, what expression is that protein or gene, kindly explain?

Answer: We modified the legends of figure 5 and 6 to specify that data are about protein expression levels in the total homogenate obtained from hippocampal tissue.

5)    This manuscript needs serious revision, the authors need to give separate section for abbreviation. Many words are irregular and not first time abbreviated. The abbreviation leads the readers in trouble to understand the results.

Answer: We checked within the text and we added the missing abbreviations in the revised form of the manuscript.

6)    The authors can also add this reference for the methods of behavioral experiments, which can merit their present study https://doi.org/10.1016/j.redox.2022.102280, https://doi.org/10.1016/j.apsb.2022.01.017

Answer: We thank the reviewer for the suggestion, we slightly modified the material and methods section. Our protocols of behavioral tests performed in rats were already published, thus we decided to cite our previous works which explained procedures, software and other details useful for the readers to replicate our findings.

7)    I request the authors to illustrate a schematic diagram to demonstrate that hippocampal alterations in NMDA (Glun2B and GluN1) and AMPA receptor subunits trafficking to the post-synapse mechanism, with conclusive evidence, for a lay man understanding to the readers.

Answer: We would like to thank the reviewer for the suggestion. In the revised form of the manuscript we added a graphical abstract in which we inserted a scheme to explain what HOMO and TIF fractions are, thus we hope that the concept of receptor trafficking inside and outside the synaptic spine would be more clear.

8)    The authors need to carefully revise the manuscript with the requested experiments and a language check would be better. Even though the authors describe those hippocampal alterations in NMDA (Glun2B and GluN1) and AMPA receptor subunits trafficking to the post-synapse mechanism with conclusive evidence. There is scarce evidence for the current hypothesis and understanding, the authors can majorly revise the manuscript and resubmit for further review. In current form the manuscript cannot be published in Nutrients.

Answer: We performed a language check, but no additional experiments were requested by the reviewers.

Reviewer 3 Report

In their interesting and important study, the authors set up a rat model for selenium restriction. In this model they linked behavioral abnormalities with sex-specific deregulation of the corresponding subunits of NMDA and AMPA receptors.

I have just minor comments:

Abbreviation of PND need to be deciphered in the abstract for clarity.

On figure 4, does symbol $$ mean p < 0.01 if so it should be indicated in the legend. Also I do not see symbol $. Is it somehow did not indicate on the figure?

Author Response

According to Reviewer #3

I have just minor comments:

Abbreviation of PND need to be deciphered in the abstract for clarity.

On figure 4, does symbol $$ mean p < 0.01 if so it should be indicated in the legend. Also I do not see symbol $. Is it somehow did not indicate on the figure?

Answer: We apologize for this mistake, due to an error in formatting the final version of the manuscript. We thus added in the revised text an $ and corrected the value of p < 0.01 in Figure 4 and relative legend. We also added in the abstract the explanation of PND abbreviation for clarity.